# Autoantibody–Abzymes with Catalase Activity in Experimental Autoimmune Encephalomyelitis Mice

**DOI:** 10.3390/molecules28031330

**Published:** 2023-01-30

**Authors:** Andrey E. Urusov, Anna S. Tolmacheva, Kseniya S. Aulova, Georgy A. Nevinsky

**Affiliations:** Institute of Chemical Biology and Fundamental Medicine, SB of the Russian Academy of Sciences, Lavrentiev Ave., 8, Novosibirsk 630090, Russia

**Keywords:** EAE model, C57BL/6 mice, catalytic antibodies, catalase activity

## Abstract

The exact mechanisms of the evolution of multiple sclerosis are still unknown. At the same time, the development in C57BL/6 mice of experimental autoimmune encephalomyelitis (EAE, simulating human multiple sclerosis) happens as a result of the violation of bone marrow hematopoietic stem cell differentiation profiles integrated with the production of toxic auto-antibodies splitting the basic myelin protein, myelin oligodendrocyte glycoprotein (MOG), histones, and DNA. It has been shown that IgGs from the plasma of healthy humans and autoimmune patients oxidize many different compounds due to their peroxidase (H_2_O_2_-dependent) and oxidoreductase (H_2_O_2_-independent) activities. Here, we first analyzed the changes in the relative catalase activity of IgGs from C57BL/6 mice blood plasma over time at different stages of the EAE development (onset, acute, and remission phases). It was shown that the catalase activity of IgGs of 3-month-old mice is, on average, relatively low (*k*_cat_ = 40.7 min^−1^), but it increases during 60 days of spontaneous development of EAE 57.4-fold (*k*_cat_ = 2.3 × 10^3^ min^−1^). The catalase activity of antibodies increases by a factor of 57.4 by 20 days after the immunization of mice with MOG (*k*_cat_ = 2.3 × 10^3^ min^−1^), corresponding to the acute phase of EAE development, and 52.7–fold by 60 days after the treatment of mice with a DNA–histone complex (*k*_cat_ = 2.1 × 10^3^ min^−1^). It is the acceleration of the EAE development after the treatment of mice with MOG that leads to the increased production of lymphocytes synthesizing antibodies with catalase activity. All data show that the IgGs’ catalase activity can play an essential role in reducing the H_2_O_2_ concentration and protecting mice from oxidative stress.

## 1. Introduction

Multiple sclerosis (MS) is a demyelinating chronic disease of the central nervous system (CNS). The etiology of MS is unclear, and the widely accepted theory of the pathogenesis of MS involves a leading role in myelin destruction caused by the inflammation associated with autoimmune reactions [1]. Several different experimental autoimmune encephalomyelitis (EAE) models mimic a specific facet of human MS (for reviews, see [2,3,4]), including C57BL/6 mice. The development of EAE in C57BL/6 mice has a spontaneous chronic–progressive course. Autoimmune diseases were first proposed as being conditioned by bone marrow hematopoietic stem cell (HSC) defects [5]. The spontaneous and antigen-induced evolution of systemic lupus erythematosus (SLE) in SLE-prone MRL-lpr/lpr mice [6,7,8] and EAE in EAE- C57BL/6 mice [9,10,11] was proven later to be a consequence of the reorganization of specific bone marrow HSCs. In autoimmune diseases (AIDs), the immune system disturbances and defects consist of particular or specific changes in the bone marrow HSCs’ differentiation profiles in parallel with the production of specific catalytic antibody–abzymes hydrolyzing DNA, RNA, polysaccharides, peptides, and proteins [6,7,8,9,10,11]. 

Antibodies with catalytic activities (antibody–enzymes; abzymes or Abzs) against transition states of various chemical reactions that catalyze more than 150 different reactions are novel important enzymes of biological fluids (for reviews, see [12,13,14,15]). Natural catalytic abzymes from the sera of blood-degrading polysaccharides, DNA, RNA, oligopeptides, and proteins were revealed in patients with many AIDs (for reviews, see [16,17,18,19,20]). With some exceptions, abzymes with these catalytic activities in conditionally healthy volunteers are absent or usually have shallow activities. The auto-antibody–abzymes with enzymatic activities were revealed as being statistically significant and the earliest markers of development of many AIDs [15,16,17,18,19,20].

Abzymes with DNase activity in SLE [21] and MS patients [22] are harmful and cytotoxic, since they induce cell apoptosis, which accelerates the development of autoimmune pathologies. In MS and SLE patients, Abzs against myelin oligodendrocyte glycoprotein (MOG) and myelin basic protein (MBP) possessing protease activity may attack proteins of the myelin–proteolipid sheath of axons. Consequently, these auto-abzymes may also play an essential detrimental role in MS pathogenesis.

The partially reduced species of oxygen (O_2_^•−^, H_2_O_2_, and OH^•^) in many organisms are produced as intermediates of aerobic respiration [23,24,25]. They can be formed in organisms through exposure to ionizing radiation. Such compounds are potent oxidants attacking various cellular DNA forms, proteins, and lipids. [23,24,25]. The oxidative damage of cell compounds is considered essential in aging, mutagenesis, and carcinogenesis. 

Some canonical antioxidant enzymes (superoxide dismutases, catalases, peroxidases, and glutathione peroxidases) provide fundamental defense mechanisms for protecting cell and blood components from oxidative modifications [25,26,27]. Mammalian antioxidant enzymes, however, are present substantially in different cells. However, only low catalytic activity levels of canonical enzymes can be revealed in the blood, since they lose their activity in the blood relatively fast [28,29]. In contrast to classical antioxidant enzymes, different antibodies can exist in the blood for a long time (1–3 months) [30]. 

It was fascinating to see whether blood catalytic antibody–abzymes could participate in protection from the oxidative stress of humans and animals. It was first demonstrated that IgG antibodies from blood plasma samples of healthy Wistar rats possess H_2_O_2_-dependent peroxidase and peroxide-independent oxidoreductase activities in the absence of H_2_O_2_. Similarly to horseradish peroxidase, the IgG antibodies of rats could effectively oxidize several different compounds, including 3,3′-diaminobenzidine (DAB) and other compounds [31,32,33,34,35]. Moreover, the IgGs of Wistar rats possess superoxide dismutase and catalase activities [36]. Then, it was shown that H_2_O_2_-dependent peroxidase and H_2_O_2_-independent oxidoreductase activities are intrinsic properties of IgGs in conditionally healthy humans and animals [31,32,33,34,35,36,37,38,39]. These redox activities of IgGs from blood plasma samples of healthy humans and patients with SLE and MS were compared [39]. The average oxidoreductase and peroxidase activities of SLE IgGs were significantly higher in comparison with Abzs from healthy humans. The catalytic activities of IgGs from patients with MS were shown to be more heightened than those from healthy donors but were lower than for SLE abzymes [39]. The obtained data indicate that abzymes with redox activities can serve as an additional blood factor for detoxifying reactive oxygen species. The protection of MS and SLE patients from some toxic compounds may be better than for healthy people [39]. 

It was interesting to find out at what stages of AID development the activation of the production of Abzs with peroxidase and oxidoreductase activities can occur. Therefore, we used three EAE-prone mouse models of MS (C57BL/6, Th, and 2D2), in which it was possible to analyze changes in the activity levels of IgGs during the onset, acute phase, and remission of this pathology [40,41,42]. On average, the peroxidase activity of mice IgGs in the oxidation of several substrates was significantly higher than the oxidoreductase activity. The relative peroxidase and oxidoreductase activities of the IgGs increased during 40 days of spontaneous development of EAE in C57BL/6, Th, and 2D2 mice. The acceleration of the EAE evolution after the mice were immunized with MOG and DNA complexes with histones, the activity levels increased much faster [40,41,42]. 

As noted above, the catalase activity of the antibodies was found in the blood plasma of Wistar rats [36].

It seemed important to understand whether the blood of the mice contained antibodies with catalase activity. In addition, it was interesting how the relative activity of the antibodies in the decomposition of hydrogen peroxide changed during the spontaneous and antigen-induced development of EAE in mice. With this in mind, we analyzed the catalase activity of IgGs from the blood plasma of C57BL/6 mice corresponding to spontaneous, MOG, and DNA–histone complex groups, accelerating the development of EAE.

## 2. Results 

### 2.1. Experimental Groups of Mice

As mentioned above, the development of EAE leads to the specific reorganization of the immune system in C57BL/6 mice, associated with significant changes in the differentiation of mouse HSCs. In parallel, there are increases in the proliferation of lymphocytes in different organs [9,10,11]. All of these defects and changes in the immune system lead to increased proteinuria and the generation of catalytically active Abs hydrolyzing DNA, MBP, and MOG [9,10,11]. As noted above, the development of autoimmune diseases in humans and animals leads to an increased abzyme concentration with enzymatic redox activities [34,35,36,37,38,39,40,41,42]. It was interesting to see how these defects and changes in the immune system can lead to possible alterations in the antibodies’ relative activities in the oxidation of various substrates. Additionally, it seemed important to compare the overtime patterns of changes in the relative activity levels of Abs with oxidative functions with those for abzymes that hydrolyze MBP, MOG, and DNA during mouse EAE development [9,10,11]. 

In the study of oxidoreductase activities, we have used homogeneous IgG preparations containing no canonical enzymes before and after the immunization of C57BL/6 mice with MOG [9,10] and a DNA–histone complex [11]; these preparations were obtained and described in [40,41,42]. 

To demonstrate the violation of the immune status in C57BL/6 mice, the integral Appendix A indicate changes over time in a number of mouse bone marrow BFU-E, CFU-E, CFU-GM, and CFU-GEMM colony-forming units (Appendix A); the relative amounts of lymphocytes in the bone marrow, spleen, thymus, and lymph nodes (Appendix A) in untreated mice; as well as after their treatment with a DNA–histone complex and MOG as described in [9,10,11,40,41,42].

It was shown that C57BL/6 mice are characterized by the very slow, spontaneous, and MOG-induced development of EAE [3,4,17,18,19,20]. Some typical indicators of EAE development (optic neuritis and other clinical or histological evidence) appear in C57BL/6 mice only 1–2 years after the spontaneous or MOG-accelerated evolution of EAE [3,4,17,18,19,20]. The appearance of auto-Abs hydrolyzing DNA, proteins, and oligosaccharides was revealed as the earliest and most statistically significant and important marker of the beginning of many autoimmune diseases in humans and mice prone to AIDs (for reviews, see [17,18,19,20]). The enzymatic activities of abzymes are veraciously detectable before the appearance of typical known medical and biochemical markers of different AIDs at the pre-disease stage [6,7,8,9,10,11,17,18,19,20]. At the pre-disease stage and with the onset of different AIDs, the concentrations of different auto-Abs usually correspond to the index spans, which are typical for healthy humans and experimental mice. The emergence of abzymes may authentically testify to the beginning of AIDs, while the increase in their enzymatic activities are coupled with the development of deep pathologies [6,7,8,9,10,11,17,18,19,20]. In this work, we analyzed the changes in the catalase activity of antibodies at the early stages of the development of EAE in C57BL/6 mice.

### 2.2. Catalase Activity

The antibodies used in this work were obtained from the blood plasma samples of C57BL/6 mice as previously described in [10,11,40,41,42]. These antibodies were electrophoretically homogeneous and were active in the hydrolysis of DNA, MBP, and histones [10,11], as well as the oxidation of 3,3′-diaminobenzidine (DAB), 2,2′-azino-bis(3-ethylbenzothiazoline-6-sulfonic acid) diammonium salt (ABTS), and some other substrates [40,41,42]. It was interesting to see whether these antibodies exhibit catalase activity.

An analysis of the catalase activity of the IgGs was carried out by reducing the absorption of hydrogen peroxide at 240 nm, according to [43]. Figure 1 shows three typical kinetic curves of H_2_O_2_ decomposing in the presence of three different IgG preparations. Such kinetic curves were obtained for each individual IgG preparation corresponding to 7 mice from different groups before and after their immunization with MOG or a DNA–histone complex (see below). All IgG preparations degraded the hydrogen peroxide but at different rates. 

Using the analysis of the activities of IgGs in the hydrolysis of DNA, MBP, MOG, and histones, as well as the peroxidase oxidation of DAB and ABTS after the SDS-PAGE of the antibodies, it was shown that all of these activities are properties of IgGs [10,11,40,41,42]. To prove that the catalase activity belongs directly to the antibodies and not to any admixtures of canonical catalases, we used an equimolar mixture of 21 IgG preparations (IgG_mix_). After the SDS-PAGE and removal of SDS, the gel was cut into small fragments, and all components of these fragments were eluted from the gel. The eluates of the gel fragments were used to analyze the contents of proteins capable of decomposing hydrogen peroxide (Figure 2). The catalase activity was detected only in the gel fragments corresponding to 150 kDa IgG_mix_. Since SDS dissociates all protein complexes, seeing the catalase activity only in the gel fragment of intact IgG_mix_ and the absence of any other protein bands and catalase activity in other fragments provides direct evidence that IgG_mix_ possesses intrinsic catalase activity (Figure 2).

### 2.3. Changes over Time in the Catalase Activity during the Development of EAE

The relative catalase activity was determined to evaluate a possible change in the activity of individual IgGs over the period of EAE development. For a further analysis, the averaged data for the apparent *k*_cat_ values (7 mice in each group) were used. 

Figure 3 demonstrates the changes in average *k*_cat_ values of Abs during the spontaneous development of EAE. 

At 50 days of age, the blood of C57BL/6 mice contains IgGs that are active in the decomposition of hydrogen peroxide (*k*_cat_ = 1.1 × 10^3^ min^−1^). The spontaneous development of EAE over 42 days from 50 to 92 days after birth (3-month-old mice; upper scale) leads to a relatively gradual decrease in catalase activity by a factor ~27.4 (*k*_cat_ = 40.7 min^−1^; *p* < 0.05). Then, starting from 92 days of life (zero time, corresponding to the beginning of experiments involving the immunization of mice with MOG and the DNA–histone complex; lower scale) to 152 days of life, there is an increase in IgG catalase activity by 57.4-fold compared to zero time (*k*_cat_ = 2.3 × 10^3^ min^−1^; *p* < 0.05). 

Three-month-old mice were used in the experiments involving the immunization of mice [9,10,11]. According to [2,3,4], the period 6–7 days after the immunization of C57BL/6 mice with MOG corresponds to the onset, the period 19–20 days after immunization corresponds to the acute stage of EAE development, while the remission phase begins after about 25–30 days. In the process of the spontaneous development of EAE, the increase in the activity of abzymes in the hydrolysis of DNA, MBP, and MOG occurs gradually and almost linearly (Appendix A) [9]. At ~7 days after the immunization of mice with MOG in parallel with the change in the HSC differentiation profile, significant increases in the MOG-, MBP-, and DNA-hydrolyzing activities of IgGs are observed. In comparison, by 20 days, the activity increases are maximal (Appendix A) [9]. These activities sharply decrease during the period of remission (>25–30 days).

In contrast to the treatment of mice with MOG [9], after the immunization of mice with the DNA–histone complex, two peaks of an increase in the DNase activity of antibodies are observed at 20 days, then the activity begins to grow rapidly after 20–25 days and up to 60 days [10,11]. Taking this into account, in this work, IgG preparations were used 20 days after the immunization of mice with MOG and 20 and 60 days after their treatment with the DNA–histone complex.

Similar to DNase-, MBP-, and MOG-hydrolyzing activities during the spontaneous development of EAE, the relative efficiency of DAB and ABTS oxidation increases almost gradually up to 40 days (Appendix A). After the immunization of mice with MOG, the peroxidase activity of IgGs also increases relatively smoothly, but by 40 days it is 3.1 times higher than after the spontaneous development of EAE [40,41,42]. The treatment of mice with DNA–histone complexes leads to a sharp increase in the peroxidase activity of antibodies by 7–10 days after immunization, followed by a slow and weak change (Appendix A). The relative peroxidase activity after the immunization of mice with MOG is about two times higher than after their treatment with the DNA–histone complex [40,41,42].

Figure 4 shows data on the relative catalase activity of IgGs before and after the treatment of mice with MOG and a DNA–histone complex. 

Whereas spontaneous development by day 20 after time zero results in a ~17.1-fold increase in catalase activity (*k*_cat_ = 695.7 min^−1^; *p* < 0.05), by day 20 after the treatment of mice with MOG, it increases by a factor of ~57.4 (*k*_cat_ = 2.3 × 10^3^ min^−1^; *p* < 0.05; Figure 4). Interestingly, the spontaneous development of EAE for 60 days leads to approximately the same increase in catalase activity (57.4 times) as the immunization of mice with MOG (20 days) (*p* > 0.05; Figure 4). Approximately the same increase in catalase activity (54.1-fold) is observed by 60 days after the immunization of mice with the DNA–histone complex (*k*_cat_ = 2.1 × 10^3^ min^−1^; *p* > 0.05). In addition, the catalase activity by 60 days is 3.4 times higher than after 20 days of spontaneous development of EAE. Thus, the immunization of mice with MOG significantly accelerates the appearance of abzymes with catalase activity compared to the DNA–histone complexes.

## 3. Discussion

It is known that reactive nitrogen and oxygen species (ROS) play an essential role in the oxidative stress processes [44]. Oxidative disorders are integral to the pathological processes in many autoimmune diseases [44,45,46,47,48,49], including MS [50,51]. ROS and subsequent oxidative damages could contribute to multiple sclerosis development by acting on particular pathological processes [52]. ROS initiates different lesions due to inducing blood–brain barrier disruption, reinforcing myelin phagocytosis and affecting leukocyte migration. They contribute to lesion persistence by mediating cellular damage and various biological macromolecules important for CNS cell function [44,45,46,47,48,49,50,51,52].

Endogenous enzymes with antioxidant activities usually counteract oxidative stress. In MS, such enzymes include overexpressed catalase, superoxide dismutases, and heme oxygenase 1 [52]. At the same time, antioxidant enzymes are primarily present in various cells, while their activity levels in the blood are usually low because they quickly lose their catalytic activity [28,29]. Some immunoglobulins are stable blood molecules, which remain intact for several months [30]. In the blood of healthy donors and rats, antibody–abzymes were revealed with antioxidant peroxidase and oxidoreductase activities [31,32,33,34,35,36,37,38]. Moreover, it was shown that the enzymatic activity levels of Abzs in the blood of patients with SLE and MS are significantly higher than in healthy donors [39]. However, the presence of abzymes with catalase activity in the blood has been shown to date only for healthy Wistar rats [36]. 

In this study, it was first shown that spontaneous EAE leads to a relatively gradual increase in the activity of antibodies with catalase activity, which is to some extent similar to the growth in enzymatic activities of IgGs hydrolyzing DNA, MBP, and MOG [9,10,11]. The immunization of mice with MOG leads to the accelerated production of abzymes decomposing H_2_O_2_ compared to mice treated with DNA–histone complexes (Figure 4). 

Relatively low reaction rates usually characterize catalysis by artificial Abzs against stable analogs of transition states; the *k*_cat_ values for such Abzs are approximately 10^2^–10^6^-fold lower as compared to canonical enzymes with the same catalytic activity [12,13,14,15]. A similar situation was observed for natural Abzs from autoimmune patients [15,16,17,18,19,20]. For example, the *k*_cat_ value for SLE DNase IgGs is 14 min^−1^ [53], the range for SLE RNase IgGs in the hydrolysis of oligonucleotides 0.01–5.8 min^−1^ [54], and the value for asthma IgGs in the hydrolysis of vasoactive peptides is 15.6 min^−1^ [55]. There is only one example of autoimmune Abzs with a specific catalytic activity comparable with that of canonical enzymes with the same activity; IgGs with RNase activity in some patients with MS could demonstrate specific activity levels ranging from 0.1 to 40% of the pancreatic RNase [56].

Unlike auto-abzymes with hydrolyzing activities, which are found mainly in the blood of patients with various AIDs, antibodies with redox functions are present in the biological fluids of healthy people and animals [15,16,17,18,19,20]. A feature of abzymes with redox functions is their high activity compared to other abzymes. For example, the *k*_cat_ values for IgGs with peroxidase activity from healthy Wistar rats varied from 1.8 × 10^2^ to 2.9 × 10^3^ min^−1^ [31,32,33]. The average relative activity of IgG preparations with peroxidase activity from 3-month-old C57BL/6 mice in the oxidation of ABTS before their immunization was *k*_cat_ = 69.8 min^−1^, which increased to 246.0 and 372.0 min^−1^ after mouse immunization with the DNA–histone complex and MOG, respectively [40,41]. In addition, the relative activity levels of Abzs in the oxidation of ABTS from the blood samples of mice and humans are high and to some extent comparable [40,41]. 

The *k*_cat_ values for classical catalases vary in the range of 2.9 × 10^4^–9.5 × 10^5^ min^−1^ [57]. The catalase activity in the blood of 3-month-old mice C57BL/6 mice is relatively low at *k*_cat_ = 40.7 min^−1^. However, the spontaneous development of EAE within 60 days leads to an increase in *k*_cat_ by 57.4 times up to ~(2.3 ± 0.2) × 10^3^ min^−1^, which is close to the values of *k*_cat_ after immunization with MOG (~2.3 ± 0.2) × 10^3^ min^−1^) and DNA–histone complexes (~2.1 ± 0.2) × 10^3^ min^−1^) (Figure 4). Thus, the redox antibodies of C57BL/6 mice with peroxidase and catalase activity are characterized by relatively high rate constants.

The decrease in catalase activity from 50 days (*k*_cat_ = 1.1 × 10^3^ min^−1^) to three months of life (*k*_cat_ = 40.7 min^−1^) by a factor of 24.5 (Figure 3) was somewhat unexpected. To analyze this phenomenon, it should be noted that a clear indicator of the development of spontaneous EAE in C57BL/6 mice is the appearance in the blood of Abs that hydrolyze DNA, MOG, and MBP. However, this occurs only in mice at about 3 months of life. Given this, at up to three months of age, C57BL/6 mice may need to be considered conditionally healthy mice. It is possible that the concentration of antibodies with catalase activity in the blood of young mice should be increased and that their maturation can lead to a decrease in their concentration. However, the spontaneous development of EAE leads to a change in the differentiation profile of the bone marrow stem cells, and as a result an increase in the concentration of abzymes with catalase activity may occur.

In the case of many antigens, including DNAs, RNAs, oligosaccharides, proteins, and lipids, a possible method of producing Abs and Abzs to these antigens is evident and well described [15,16,17,18,19,20]. The origin of Ab–Abzs with peroxidase and catalase functions is not yet clear. It is theoretically possible to assume several origins. At first, Abzs with different catalytic redox activities in healthy humans and patients with autoimmune diseases could reflect the constitutive synthesis of germline antibodies described by Paul’s group [58,59]. Abzymes with peroxidase activity production could be stimulated by different toxic, mutagenic, and carcinogenic compounds, forming complexes with some proteins, which can result in the production of Abzs against these haptens. At the same time, it is known that the second anti-idiotypic antibodies against various enzymes’ catalytic sites also possess catalytic activities [15,16,17,18,19,20]. Thus, in principle, one cannot exclude the possibility that different redox abzymes can be anti-idiotypic antibodies against canonical catalases, peroxidases, glutathione peroxidases, superoxide dismutases, and other enzymes oxidizing different compounds. 

Considering the data given above, the question arises of why Abs with several redox functions are needed. The predominance of pro-oxidant processes in the antioxidant system and insufficiency can lead to the impaired hypofunction of NMDA receptor myelination, the death of interneurons, and other different abnormal processes, thereby contributing to the development of pathology [48,49]. 

Abzymes from healthy donors, animals, and patients with AIDs oxidizing different toxic compounds harmful to humans can protect the body from oxidative stress. Due to prolonged circulation and the high content in the blood, such IgGs can decrease oxidative disorders in the bloodstream. Moreover, due to the ability of Abs to accumulate in the foci of inflammation, catalytic IgG–abzymes with catalase and peroxidase activities, along with classical catalases and glutathione peroxidases [60], can regulate the hydrogen peroxide concentration and limit the damage caused by the high concentration of H_2_O_2_ and oxidative processes in these specific zones.

The biological significance of Abzs with redox activities in humans and animals could be more essential than it might seem at first glance. It was shown that different polyclonal and monoclonal Abs from different mammals possess superoxide dismutase activity and efficiently reduce singlet oxygen (^1^O_2_^*^) to ^●^O_2_^–^, leading to H_2_O_3_ as the first intermediate, while a particular cascade of chemical reactions finally leads to the formation of H_2_O_2_ [61,62]. The authors supposed that the origin of this enzymatic activity is based on the adaptive maturation of the antibody’s variable domains. These data indicate a possible protective function of these Abzs and raise the question of whether the detoxification of ^1^O_2_^*^ could play an important role in the evolution of Abs. These abzymes show the possible mechanisms of how oxygen could be reduced and recycled via phagocyte action, leading to an increase in the antimicrobicidal action of the immune system [61,62]. Thus, it seems likely that in healthy organisms a specific set of Abzs exists with several redox-protective functions. Some abzymes with superoxide dismutase activity could convert oxygen into H_2_O_2_, while other abzymes with catalase and peroxidase activities could neutralize hydrogen peroxide. Because many various toxic mammalian compounds may be substrates of catalase, peroxidase, and oxidoreductase IgGs [31,32,33,34,35,36,37,38,39,40,41,42], it is easy to suppose that a set of possible substrates of catalytic Abs can be extremely wide. Taken together, polyclonal Abzs with different redox functions can serve as an additional system of detoxification of reactive oxygen species in humans and different animals. 

## 4. Materials and Methods

### 4.1. Reagents 

All chemicals, Superdex 200 HR 10/30 (17-5175-01), Protein A-Sepharose (17046901), Protein G-Sepharose (17061801), and columns were obtained from GE Healthcare Life Sciences (New York, NY, USA). These preparations were free from possible contaminants. The MOG_35–55_ (2568/1) was obtained from EZBiolab (Munich, Germany), while the Bordetella pertussis toxin (*M. tuberculosis*; BML-g100-0050) was obtained from the Native Antigen Company (Oxfordshire, UK).

### 4.2. Experimental Animals 

The C57BL/6 mice (3-month-old) were grown in a vivarium for mice using special conditions free of any viral, bacterial, or other pathogens at the Institute of Cytology and Genetics (ICG) [9,10,11,40,41]. All experiments and analyses were performed by us in accordance with the ICG bioethical committee’s protocols according to known principles for working with animals based on the European Communities Council Directive 86/609/CEE. The bioethical committee of the Institute of Cytology and Genetics supported our study. To analyze the changes in catalase activity of IgGs during the spontaneous development of EAE, C57BL/6 mice from 50 to 152 days after birth were used. Immunization experiments were carried out using three-month-old mice (92 days of life; zero time).

### 4.3. Immunization of Mice 

In this study, we used IgG preparations that had been utilized earlier for the analysis of different parameters characterizing the development of EAE in C57BL/6 mice before and after their immunization with MOG and complexes of DNA with five histones (H1, H2A, H2B, H3, and H4; DNA–histone complexes) [9,10,11]. In this study, we used several groups of 7 mice as described in [9,10,11]. One of the groups was subjected to standard decapitation and blood sampling at time zero (3-month-old mice); the other two groups were decapitated 20 and 60 days after the spontaneous evolution of EAE. One of the groups was used for an analysis 20 days after immunization with MOG at zero time, and two groups 20 and 60 days after immunization at zero time with the DNA–histone complex.

For the purification of IgGs, 0.7–1 mL of blood was collected 7–60 days after treatment by decapitation using standard approaches [9,10,11]. The methods used for the immunization of mice have been published earlier [9,10,11], and their more detailed descriptions are given in the Appendix A

### 4.4. IgG Purification

The electrophoretically and immunologically homogeneous IgG preparations were isolated as in [40,41,42] via the chromatography of blood plasma proteins on Protein G-Sepharose and then using FPLC gel filtration in severe conditions (pH 2.6). For the protection of IgGs from possible contamination, the IgG samples were filtered through special filters (0.1 μm). The solution aliquots were kept at −70 °C before use in different analyses. The SDS-PAGE assay of the IgGs for homogeneity was performed using 4–15% gradient gels as in [40,41,42], and the proteins were visualized via silver staining (more detailed data on the purification of IgGs are given in the Appendix A).

To exclude possible artifacts due to hypothetical traces of contaminating catalases, the IgGs were separated using SDS-PAGE. Their catalase activities were detected using an SDS-PAGE assay as in [40,41,42]. After the use of SDS-PAGE to restore the catalase activity, the SDS was removed via gel incubation for 3 h at 23 °C with K-phosphate (pH 7.0) and washed 14 times with this buffer. The gel strips were cut into small pieces (2.6–3.1 mm), which were carefully crushed and placed in tubes containing 50 μL of K-phosphate (pH 7.0) buffer. For 10 days, the tubes were kept at 4 °C with periodic shaking using a mixer. Then, the gel was removed by centrifugation and the solutions were used to determine the catalase activity. The parallel longitudinal gel strips were used to detect the IgGs’ position on the gel via silver staining. This activity was revealed only in the band of intact IgGs, and there were no other peaks of proteins or catalase activity.

### 4.5. Assay of Catalase Activity

The relative catalase activity of the IgGs was estimated using optimal conditions [36]. The catalase activity assay was carried out using an analysis of the H_2_O_2_ decomposition by measuring the reaction mixture’s absorption at 240 nm according to [43]. The reaction mixture (100 µL) contained 50 mM of phosphate buffer (pH 7.0), 30 mM of H_2_O_2_, and 0.04–0.1 mg/mL of IgGs. The reaction mixture was incubated at 25 °C in a spectrophotometric cuvette (path length 1 cm) for 0–5 min, and the amount of product was measured every 2 s using a Genesys 10S Bio UV/Vis (Thermo Scientific, New York, NY, USA) UV–visible spectrophotometer. Reaction mixtures containing no antibodies were used as controls. The apparent rate constants (*k*_cat_) were estimated using the difference (∆A_240_) in the reaction mixture absorbance at 240 nm corresponding to the linear parts of the kinetic curves. The concentration of IgG-dependent decomposed H_2_O_2_ was estimated using a molar absorption coefficient of H_2_O_2_ of ε = 0.081 mM^−1^ sm^−1^ [63]. The *k*_cat_ values were calculated using the equation *k*_cat_ = V (M/min)/[IgG] (M).

### 4.6. Statistical Analysis

The results are reported as the means and standard deviations of at least 2–3 independent experiments for each IgG preparation, averaged over 7 different mice in every group. The significance of differences (*p*) between groups was calculated using the Mann–Whitney test.

## 5. Conclusions

Taken together, autoimmune-prone C57BL/6 mice are characterized by the spontaneous and accelerated development of EAE after their immunization with MOG and DNA–histone complexes associated with changes in the differentiation of bone marrow HSCs and the production of abzymes hydrolyzing DNA, RNA, MBP, MOG, and histones. The specific infringement of the immune status of C57BL/6 mice during the development of EAE also leads to increases in the activity of autoantibodies with peroxidase and oxidoreductase, as shown in this article, as well as the catalase activities of IgG antibodies. Thus, redox abzymes with increased activity in SLE and MS patients and various mice prone to autoimmune diseases can protect them from some toxic compounds somewhat better than in healthy people and animals.

## Figures and Tables

**Figure 1 molecules-28-01330-f001:**
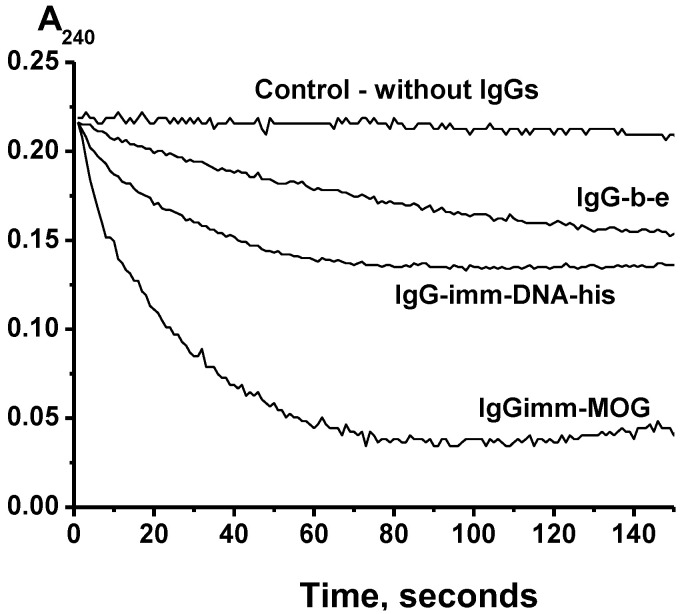
Typical examples of the time-dependences of H_2_O_2_ decomposing in the presence of three different IgG preparations (0.1 mg/mL). All designations are given in the panel; IgG-b-e is one of preparations corresponding to the group of 3-month-old mice at the beginning of the experiment; IgG2-imm-DNA-his is one of the preparations corresponding to the group of mice immunized with the DNA–histone complex; IgG-imm-MOG corresponds to mice immunized with MOG (see below).

**Figure 2 molecules-28-01330-f002:**
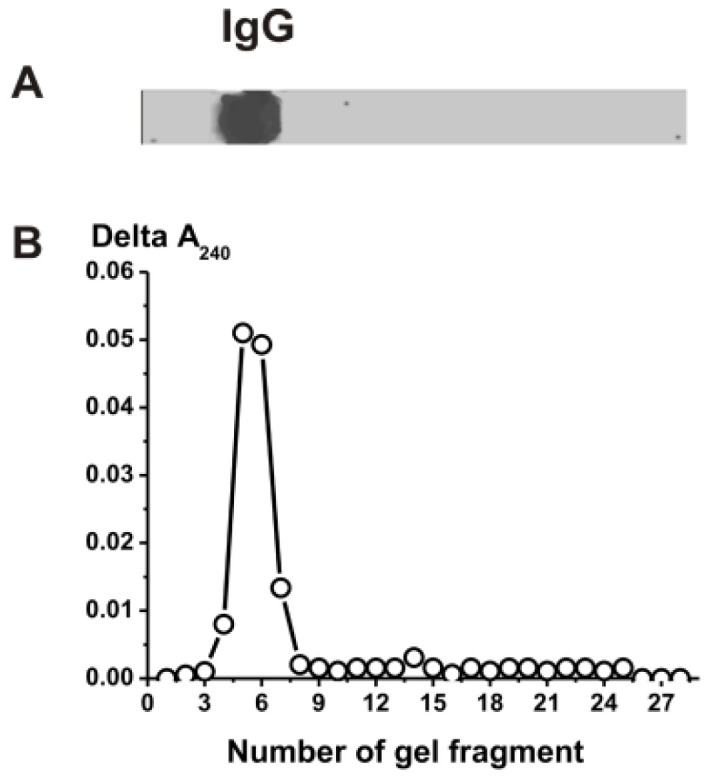
SDS-PAGE analysis of the electrophoretic homogeneity of IgG_mix_ corresponding to a mixture of 21 individual IgG preparations (19 μg) in 4–18% gradient gel followed by silver staining (**A**). After SDS-PAGE, the gel was incubated using the special solution to remove the SDS and for IgG renaturation. The relative catalase activity was estimated using 15 µL extracts of 2.6–3.1-mm fragments of one longitudinal gel slice (**B**). The errors from 2 independent experiments in the initial rate determination process did not exceed 8–10%.

**Figure 3 molecules-28-01330-f003:**
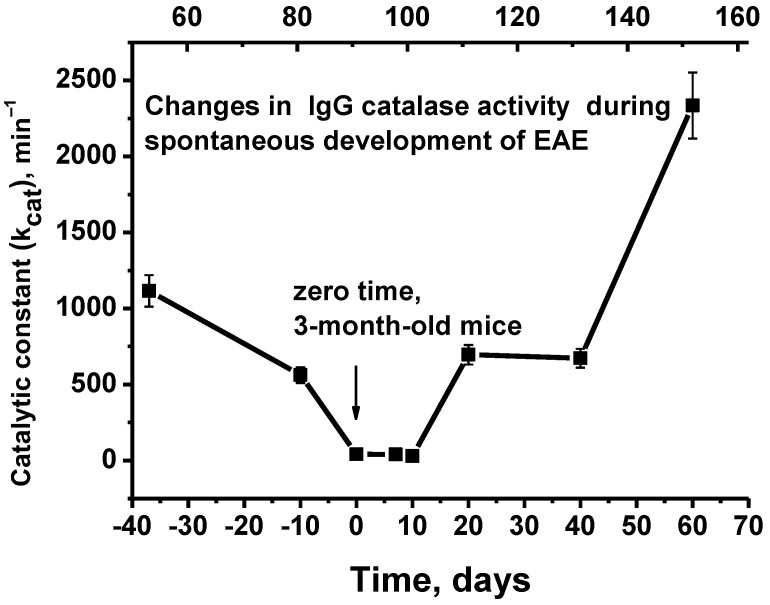
Changes over time of the average catalase activity of IgGs (7 mice in each group) during the spontaneous development of EAE before immunization in mice. The upper scale corresponds to the total number of days of the experiments, while the lower scale indicates the beginning of the experiments (zero time; 3-month-old mice), with the mice being immunized with MOG and the DNA–histone complex. The changes in catalase activity after immunization are shown in Figure 4.

**Figure 4 molecules-28-01330-f004:**
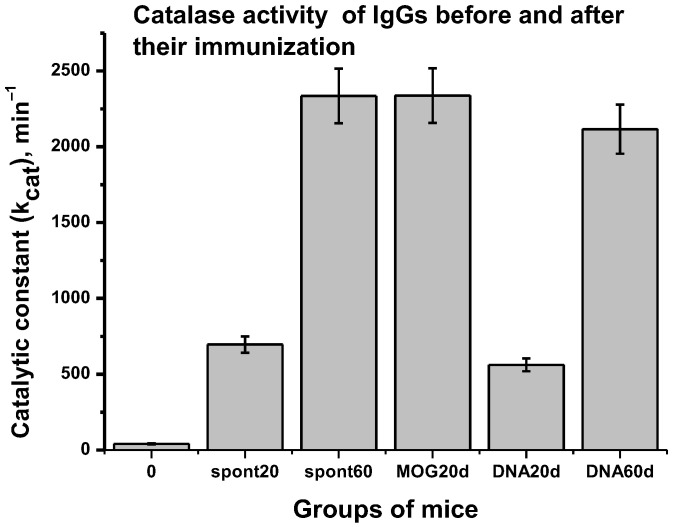
Relative average catalase activity of IgGs from different mouse groups: 0 (3-month-old mice), spon20 and spon60 (spontaneous development of EAE during 20 and 60 days, respectively), MOG20d (20 days after immunization of mice with MOG), and DNA20d and DNA60d (20 and 60 days after immunization of mice with the DNA–histone complex, respectively).

## Data Availability

All data is contained in the article and Appendix A.

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
