# Peer review of "Autoantibody–Abzymes with Catalase Activity in Experimental Autoimmune Encephalomyelitis Mice"

_molecules, 2023, doi:10.3390/molecules28031330_

Round 1

Reviewer 1 Report

The paper established a model of EAE in mice. By detecting changes catalase activity of IgGs in serum at different stages of EAE, the authors confirmed that IgGs’s catalase plays an important protective role in the oxidative stress response of mice. The results may provide theoretical reference for the treatment of autoimmune disease encephalomyelitis. However, the overall structure of this manuscript is somewhat confusing and has the following problems that need to be revised.

 1. In terms of the structure of the article, it is recommended to write the materials and methods, and then write the results. Please check the order of the article subheading, such as, line 164, is the article subheading 3.4?

 2. In Figure. 1, which three IgG preparations are represented by IgG1, IgG2 and IgG3? Please specify the name of the preparation.

 3. The ordering of references is quite confusing, please check the ordering of references carefully. 

4. Please provide a brief explanation of how the immunization cycle is determined in mice.

 5. What method is used to collect mouse blood?

 6. In Figure 4, both spont group and DNA group included 20d and 60d statistics, but why did the MOG treatment group only have 20d analysis?

Author Response

The paper established a model of EAE in mice. By detecting changes catalase activity of IgGs in serum at different stages of EAE, the authors confirmed that IgGs’s catalase plays an important protective role in the oxidative stress response of mice. The results may provide theoretical reference for the treatment of autoimmune disease encephalomyelitis. However, the overall structure of this manuscript is somewhat confusing and has the following problems that need to be revised.

  1. In terms of the structure of the article, it is recommended to write the materials and methods, and then write the results. Please check the order of the article subheading, such as, line 164, is the article subheading 3.4?

Answer: Sorry, but in the Molecules Journal the “Materials and Methods” part should be in the final after the “Discussion”. The numberings of subheadings are corrected.  

  1. In Figure. 1, which three IgG preparations are represented by IgG1, IgG2 and IgG3? Please specify the name of the preparation.

Answer: There were many preparations and curves that three preparations corresponding to different groups of mice were simply chosen.

Ig G1 one of preparation corresponding to the group of 3-month-old mice at the beginning of experiment; IgG2 – one of preparation corresponding to the group of mice immunized with DNA-histones complex, and IgG3- to mice immunized with MOG (see below).

  1. The ordering of references is quite confusing, please check the ordering of references carefully.

Answer: It was corrected

  1. Please provide a brief explanation of how the immunization cycle is determined in mice.

Answer: we have added the next information

In this study we used several groups of 7 mice described in [9-11]. One of the groups was subjected to standard decapitation and blood sampling at time zero (3-month-old mice); the other two groups were decapitated 20 and 60 days after the spontaneous evolution of EAE. One of the groups was used for analysis 20 days after mice immunization with MOG at zero time, and two groups 20 and 60 days after mice immunization at zero time with the DNA-histones complex.

  1. What method is used to collect mouse blood?

Answer: The blood of mice was collected after their standard decapitation.

  1. In Figure 4, both spont group and DNA group included 20d and 60d statistics, but why did the MOG treatment group only have 20d analysis?

Answer: As we have shown earlier [9-11], the maximum increase in the activity of abzymes is observed at 20 days (the acute phase) after mice immunization with MOG. Given this, mice were used 20 days after immunization with MOG. At the same time, after immunization of mice with the DNA-histones complex, there are two phases of an increase in the activity of abzymes, which observed - at 20 and 60 days. Therefore, an analysis of catalase activity of these mice was carried out after 20 and 60 days after immunization.

Thank you for useful remarks

Prof. Georgy A. Nevinsky

Reviewer 2 Report

If the manuscript is accepted for further revision, I recommend that the manuscript should be thoroughly re-edited in collaboration with a professional English editing service.  

Specific comments:

Introduction

p.1 / line 35: Some autoimmune diseases… ff. rephrase this sentences. The English is very bad.

p.1 / line 42: define Abs-abzymes.

p.2 / line 46: what is «blood sera»??

p2 / line 52: define DNase Abzs

Results

Figure 3: separate curves should be shown for mice with spontaneous EAE, immunized with MOG and DNA-Histone complex, respectively. As shown currently, this graph has no meaning.

P3/line 107: as mentioned above… rephrase this sentence.

P3/ line 107ff 1. Paragraph is repetition of introduction and not results.

References: the numbering of references does not correspond to the references in the text – e.g. Ref 55 has Ref 20 on page 12/line 539.

Author Response

Specific comments:

Introduction

p.1 / line 35: Some autoimmune diseases… ff. rephrase this sentences. The English is very bad.

Answer: it was corrected

p.1 / line 42: define Abs-abzymes.

Answer: it was done

p.2 / line 46: what is «blood sera»??

Answer: it was corrected

p2 / line 52: define DNase Abzs

Answer: it was corrected

Results

Figure 3: separate curves should be shown for mice with spontaneous EAE, immunized with MOG and DNA-Histone complex, respectively. As shown currently, this graph has no meaning.

Answer:

Sorry, but there are a lot of curves, and putting seven curves for each group into one figure leads to a very messy picture, because some of them overlap and it's hard to see how looks each of them. Therefore, we have given only three curves to show how they look in principle. The corrected version indicates which groups of mice these curves belong to. In order for the picture not to look dirty, it is necessary to provide a separate picture for each preparation, but this is not possible, there are many IgGs.

P3/line 107: as mentioned above… rephrase this sentence.

Answer: it was corrected

P3/ line 107ff 1. Paragraph is repetition of introduction and not results.

Answer: Sorry, but we did not understand to which part of the text this remark refers. If to the introduction to the “Results”, then this must be repeated to some extent, for the subsequent understanding of the following results.

References: the numbering of references does not correspond to the references in the text – e.g. Ref 55 has Ref 20 on page 12/line 539.

Answer:

List of references was corrected

Thank you for useful remarks

Prof. Georgy A. Nevinsky

Round 2

Reviewer 2 Report

Title: Autoantibodies-Abzymes is not correct. Abzyme is defined as Antibody with catalytic/enzymatic activity, thus it contains already the term "antibody". As in a title no abbreviations should be used, the title should be Autoantibodies with Catalase Activity in Experimental Autoimmune Encephalomyelitis Mice.

Introduction / abbreviations:

Abzymes still not defined as an abbreviation. Antibodies-abzymes… see above.

Autoantibodies with catalytic activity are auto-abzymes. Always use the same wording. Many different "expressions" are in the manuscript: Antibodies-abzymes, Abs-abzymes, catalytic abzymes, auto-Abs-abzymes, Abzs, catalytic Abs-abzymes, IgGs-abzymes,

Introduction: "AIDs development" is not correct: either AID development or development of AIDs.

Figure 1: please avoid IgG1, IgG2, IgG3, as they are specified as IgG-subclasses. Instead, lable the curves with the specific antigen, e.g. before immunization, DNA/Histone, MOG.

Figure 3: Still not clear which catalytic activity is shown after Time zero. Spontaneous EAE of non-immunized mice, MOG or DNA/histone immunized mice. If non-immunized mice are shown, the indication about immunizations is not needed. Otherwise show separate curves for spontaneous, MOG and DNA-Histone immunized mice.

Discussion: "Somewhat unexpected were the results of a decrease in catalase activity in the period from 50 (kcat = 1.1x103 min-1) to three months of mice life (kcat = 40.7 min-1) by a factor of 27.4 (Figure 3)." Should be "from 50 days to three months…"

"In the case of many antigens, including DNAs, RNAs, oligosaccharides, proteins, 301 lipids, etc.), a possible way of producing Abs…" the closing ")" should be removed.

I am still missing the correlation of catalytic activity with the clinical score of EAE, or SLE respectively.

General: improve English wording in collaboration with an English-native editor, as many phrases can be improved.   

Author Response

Abzymes still not defined as an abbreviation. Antibodies-abzymes… see above.

Autoantibodies with catalytic activity are auto-abzymes. Always use the same wording. Many different "expressions" are in the manuscript: Antibodies-abzymes, Abs-abzymes, catalytic abzymes, auto-Abs-abzymes, Abzs, catalytic Abs-abzymes, IgGs-abzymes,

Answer: Sorry, but abzymes are not always auto-antibodies, they can also be developed against external antigens. In order to avoid tuftology in different phrases, depending on their meaning, it is necessary to use different modifications of the words antibodies-abzymes. Taking into account your remark, an attempt was made to unify the designation of the words antibodies-abzymes.

The definition of this term is given: Antibodies with catalytic activities (antibodies-enzymes; abzymes or Abzs) against transition states of various chemical reactions. Then some variants of the term were replaced.

Introduction: "AIDs development" is not correct: either AID development or development of AIDs.

 Answer:

It was corrected

Figure 1: please avoid IgG1, IgG2, IgG3, as they are specified as IgG-subclasses. Instead, lable the curves with the specific antigen, e.g. before immunization, DNA/Histone, MOG.

Answer:

It was corrected

Figure 3: Still not clear which catalytic activity is shown after Time zero. Spontaneous EAE of non-immunized mice, MOG or DNA/histone immunized mice. If non-immunized mice are shown, the indication about immunizations is not needed. Otherwise show separate curves for spontaneous, MOG and DNA-Histone immunized mice.

Answer:

It was corrected

Discussion: "Somewhat unexpected were the results of a decrease in catalase activity in the period from 50 (kcat = 1.1x103 min-1) to three months of mice life (kcat = 40.7 min-1) by a factor of 27.4 (Figure 3)." Should be "from 50 days to three months…"

Answer:

It was corrected

"In the case of many antigens, including DNAs, RNAs, oligosaccharides, proteins, 301 lipids, etc.), a possible way of producing Abs…" the closing ")" should be removed.

 Answer:

It was corrected

I am still missing the correlation of catalytic activity with the clinical score of EAE, or SLE respectively.

Answer: At the onset of the disease - up to about a year or even longer - mice do not have pronounced clinical signs of the disease. Taking into account your remark, the following text has been added to the article:

It was shown that C57BL/6 mice are characterized by a very slow spontaneous and MOG-induced development of EAE [3,4,17-20]. Some typical indicators of EAE development (optic neuritis and other clinical or histological evidence) appear in C57BL/6 mice only 1-2 years after spontaneous or MOG-accelerated evolution of EAE [3,4,17-20]. The appearance of auto-Abs hydrolyzing DNA, proteins and oligosaccharides was revealed as the earliest and statistically significant and undoubtedly important marker of the beginning of many autoimmune diseases in humans and mice prone to AIDs (for review, see [17-20]). Enzymatic activities of abzymes are veraciously detectable before the appearance of typical known medical and biochemical markers of different AIDs at the pre-disease stage [6-11,17-20]. At the pre-disease stage and onset of different AIDs, the concentrations of different auto-Abs usually correspond to the indices spans, which are typical for healthy humans and experimental mice. The emerging of abzymes may authentically testify about the beginning of AIDs, while the increase in their enzymatic activities is coupled with the development of deep pathologies [6-11,17-20]. In this work, we analyzed the changes in the catalase activity of antibodies at the early stages of the development of EAE in C57BL/6 mice.

Thanks for your comments

Best wishes

Prof Georgy Novinsky